# Experimental and Simulation of Diesel Engine Fueled with Biodiesel with Variations in Heat Loss Model

**Daniel Romeo Kamta Legue** [1,2,*], **Zacharie Merlin Ayissi** [2,3,4,*], **Mahamat Hassane Babikir** [1], **Marcel Obounou** [1,2] and **Henri Paul Ekobena Fouda** [1]

1   Department of Physics, Energy—Electrical and Electronic Systems, University of Yaounde 1, Yaoundé P.O. Box 812, Cameroon; hassanemahamat6@gmail.com (M.H.B.); marcelobounou@yahoo.fr (M.O.); hekobena@gmail.com (H.P.E.F.)
2   Laboratory E3M, University of Douala, Douala P.O. Box 2701, Cameroon
3   Nationale Higher Polytechnique School, University of Douala, Douala P.O. Box 2701, Cameroon
4   Department of Automotive Engineering and Mécatronics, ENSPD, University of Douala, Douala P.O. Box 2701, Cameroon
*   Correspondence: campusromo@yahoo.fr (D.R.K.L.); merlin.ayissi@gmail.com (Z.M.A.)

**Abstract:** This study presents an experimental investigation and thermodynamic 0D modeling of the combustion of a compression-ignition engine, fueled by an alternative fuel based on neem biodiesel (B100) as well as conventional diesel (D100). The study highlights the effects of the engine load at 50%, 75% and 100% and the influence of the heat loss models proposed by Woschni, Eichelberg and Hohenberg on the variation in the cylinder pressure. The study shows that the heat loss through the cylinder wall is more pronounced during diffusion combustion regardless of the nature of the fuels tested and the load range required. The cylinder pressures when using B100 estimated at 89 bars are relatively higher than when using D100, about 3.3% greater under the same experimental conditions. It is also observed that the problem of the high pressure associated with the use of biodiesels in engines can be solved by optimizing the ignition delay. The net heat release rate remains roughly the same when using D100 and B100 at 100% load. At low loads, the D100 heat release rate is higher than B100. The investigation shows how wall heat losses are more pronounced in the diffusion combustion phase, relative to the premix phase, by presenting variations in the curves.

**Keywords:** biodiesel; load; heat loss; cylinder pressure; heat release rate; python

## 1. Introduction

Automobile pollution accounts for a large percentage of global pollution [1]. This pollution is largely caused by the use of fossil oil derivatives as fuel vehicles.

Spark-ignition and compression-ignition engines are the most widely used as a converter of the internal energy of fuels. They use petrol and diesel, respectively, as fuel. Both types of engines convert the internal energy of the fuels into thermal energy which is used in mechanical form to drive motor vehicles. Generally, heavy-duty vehicles, about 90% of the vehicles in use, are mostly diesel [2]. Self-ignition combustion offers better volumetric efficiency, which means that the diesel engine using this mode of combustion exploits a known energy potential. However, diesel engines present a serious problem for global health [3,4]. The intrinsic chemical composition of this fuel is largely responsible for this phenomenon. The combustion mode fossil fuels should be improved with regard to the level of pollutants emitted. Indeed, the emission of nitrogen oxides would be relatively responsible for pathologies linked to the proliferation of certain types of malignant tumors. In fact, this engine generally operates with poor and non-homogeneous mixtures, which leads to a relatively high level of nitrogen oxides ($NO_x$) and particulate matter (PM) [1,5,6]. The two major pollutants, particularly nitrogen oxides, are strongly dependent on the temperature of the combustion chamber, which is a function of the cylinder pressure resulting from the high compression ratio imposed by the performance research of this type

of engine. The mechanism of formation of these pollutants is explicitly described by the Zeldovich mechanism [1,3,7].

The combustion of diesel engines offers a wide area of investigation [4,8,9]. Experimental work is being carried out. Ducted fuel injection (DFI) is proposed in the literature as a strategy to improve the fuel/gas load mixture of the compression-ignition engine relative to conventional diesel combustion (CDC) [5–7]. The concept of DFI is to inject each fuel spray through a small tube into the combustion chamber to facilitate the creation of a leaner mixture in the self-ignition zone, compared to a fuel jet not surrounded by a duct. The experiments are interesting. Nilsen et al. [5] studied the effects on emissions and engine efficiency using a two-hole injector tip for charging gas mixtures containing 16 and 21% mol oxygen. DFI seems to confirm that it is effective in reducing engine soot emissions. Soot and NOx are reduced with increasing dilution. Christopher [6] conducted an investigation where DFI and CDC were directly compared at each operating point in the study. At the low-load condition, the intake charge dilution was swept to elucidate the soot and $NO_x$ performance of DFI. The authors mentioned that DFI likely has slightly decreased fuel conversion efficiencies relative to CDC. All these experimental studies are interesting and very promising, but they are still in the laboratory domain and require expensive instruments.

For economic reasons, research is increasingly oriented towards modeling engine combustion. The evaluation of wall heat losses, which are responsible for energy losses in thermal engines, is likely to optimize the operation of this type of engine. Modeling is based on control parameters such as rotational speed, compression ratio, engine load, combustion and injection models [7,10–12]. Modeling consists of reproducing physical phenomena using mathematical equations. This method offers a real-time saving and a significant reduction in costs. However, it is important to validate the results obtained from the modeling or simulation systems in order to give the results a formal character.

Zero-dimensional (0D) thermodynamic models of combustion are widely used in the literature. Caligiuri et al. [13] implemented a triple Wiebe model in order to describe and predict the heat release rate and ignition delay of dual-fuel combustion. The Aktar model has been successfully implemented and validated. Prakash correlation relating to the ignition delay was used. A methodological approach based on the prediction of the ignition delay of diesel engines has been implemented. Hariram et al. [14] conducted an investigation, and this experimental and theoretical study addressed the problem related to the effects of beeswax biodiesel mixtures with fossil diesel on combustion parameters. The zero-dimensional thermodynamic model was used as a numerical implementation approach. The variation in cylinder pressure, the net heat release rate and the ignition delay were addressed based on Webe's triple function. The code developed was successfully validated. These models allow evaluating the performance of thermal engines by the use of correlations based on thermodynamic laws.

The advantage of zero-dimensional thermodynamic models is that they are easy to use and do not require large computer memory capacities. The use of thermodynamic postulates makes them accessible and efficient. Thermodynamic 0D models allow the study of both fossil fuel combustion and alternative fuels.

The polluting nature of fossil fuels and the depletable nature and recurrent price fluctuations of fossil energy sources have led researchers for years to focus increasingly on alternative energy sources such as biofuels [4]. However, the production of biofuels has so far remained relatively low in relation to demand, given the many structural constraints, hence the need to maximize energy savings by optimizing the energy converters that use these fuels. Around 60% of energy in thermal engines is lost in several forms. However, engine losses remain a real problem as they account for almost 40% of the energy lost [15–17]. A numerical study of the heat loss through the wall would allow a better understanding of this phenomenon and improve the combustion and, by deduction, the efficiency of engines.

Several published articles have reported on the sensitivity between biodiesel and engine load compared to conventional diesel fuel; however, these do not show the specificity of the impact of the load on the cylinder pressure, taking into account the heat loss through the cylinder wall and the type of fuel used. This reasearch proposes a comparative study of the cylinder pressures generated by the combustion of diesel and neem biodiesel. The study is conducted taking into account the correlations between the heat loss through the cylinder wall. The heat release produced allows representing the phenomenology of the heat loss through the cylinder wall by dissociating the gross heat release rate from the net heat release rate during a combustion phase.

## 2. Modeling of Zero-Dimensional Thermodynamic Combustion

A modeling of the amount of energy released by combustion and a deduction of the cylinder pressure produced during the closed part of the cycle are proposed.

The chemical formulation of the fuel is the form $C_xH_yO_z$. The complete combustion equation of the fuel is given by Equation (1). The first principle of thermodynamics is applied in Equation (2) [18].

$$C_xH_yO_z + a(O_2 + 3.77N_2) \rightarrow xCO_2 + \frac{y}{2}H_2O + (3.77a)N_2 \tag{1}$$

$$dU = \delta W + \delta Q_W + \sum h_j dm_j \tag{2}$$

where $dU$ represents the variation in the internal energy of the system, $\delta W$ is the work supplied by the piston to the system, $\delta Q_W$ is the heat loss through the cylinder wall. The term $\sum h_j dm_j$ represents the energy due to the variation in mass. The heat loss through the cylinder wall is modeled using three characteristic postulates: Woschni, Eichelberg and Hohenberg. By transforming and simplifying Equation (2), the differential Equation (3) is obtained as a function of the crankshaft angle ($\theta$) [19].

$$\begin{cases} \frac{dp}{d\theta} = \frac{\gamma p}{V}\frac{dV}{d\theta} + \frac{(\gamma-1)}{V}\frac{dQ}{d\theta} \\ \frac{dT}{d\theta} = T(\gamma-1)\left[\frac{1}{pV}\frac{dQ}{d\theta} - \frac{1}{V}\frac{dp}{d\theta}\right] \end{cases} \tag{3}$$

where $p$ and $T$ represent the pressure and the cylinder temperature, $\gamma$ is the specific heat ratio, $\theta$ is the crankshaft angle and $V$ is the cylinder volume. The expression of this volume is governed by Equation (3) [19–21].

$$V(\theta) = \frac{\pi D^2 S}{8}\left(1 - \cos(\theta) + \lambda - \sqrt{\lambda^2 - sin^2(\theta)} + \frac{2}{CR - 1}\right) \tag{4}$$

The variation in the cylinder volume $V$ in relation to the crankshaft angle can be deduced [19].

$$\frac{dV(\theta)}{d\theta} = \frac{\pi D^2 S}{8}\left(1 - \frac{cos\theta}{\sqrt{\lambda^2 - sin^2\theta}}\right)sin\theta \tag{5}$$

where $\lambda$ is the ratio of the rod length, $D$ is the cylinder bore, $CR$ is the compression ratio and $S$ is the stroke.

The term $\frac{dQ}{d\theta}$ is the heat release rate of the system expressed as follows [19]:

$$\frac{dQ}{d\theta} = \frac{\gamma}{\gamma - 1}p\frac{dV}{d\theta} + \frac{1}{\gamma - 1}V\frac{dp}{d\theta} \tag{6}$$

By entering the experimental combustion data into Equation (6), the heat release profile in the engine can be reconstructed as recommended by many authors using the analysis model [11,14,22]. The evolutions of the heat release profile(s) for an engine speed of 1500 rpm correspond to the load ranges: 25%, 50%, 75% and 100%. These loads correspond to 1.1, 2.5, 3.3 and 4.5 kW, respectively.

Four steps make up the modeling of the heat release, which are characteristic of the different terms used: the energy lost at the walls $\frac{dQ_w}{d\theta}$ integrates the heat exchange coefficient of the three models mentioned above; the energy released by combustion takes into account the internal energy of the fuel $\frac{dQ_{comb}}{d\theta}$ and two other terms due to exchanges with the surroundings (inlet and outlet mass flow).

The system is considered closed, and both terms are taken as zero. The system of Equation (7) is proposed.

$$\begin{cases} \frac{dQ_{comb}}{d\theta} = m_{inj} * LCV * \frac{dx_b}{d\theta} \\ \frac{dQ_w}{d\theta} = h_c A(\theta)(T - T_w)\frac{1}{\omega} \end{cases} \tag{7}$$

The terms appearing in Equation (7) are defined by Equation (8) at the heat exchanger surface as a function of the engine geometry. The characteristic fraction of fuel burned as a function of the crankshaft angle variation is given by Equation (9).

$$A(\theta) = \left(\pi * \frac{D^2}{2}\right) + \pi * D * \frac{L}{2}(\lambda + 1 - Cos\theta - \sqrt{\lambda^2 - Sin^2(\theta)}) \tag{8}$$

$$x_b = 1 - exp\left[-a\left(\frac{\theta - \theta_0}{\Delta\theta}\right)^{m+1}\right] \tag{9}$$

The fraction of fuel burnt in relation to the crankshaft angle $\theta$ is evaluated. $m_{inj}$ is the mass of the injected fuel, $T_w$ is the wall temperature, $\omega$ is the engine rotation frequency, LCV is the characteristic lower calorific value of the fuel used and $h_c$ is the convective heat loss coefficient depending on the heat loss model used, expressed in kW/m$^2$K.

The different heat loss coefficients through the walls are proposed [20].

$$h_c = 3.26 * D^{-0.2} * p^{0.8} * T^{-0.55} * W^{0.8} \tag{10}$$

$$h_c = 0.013 * V^{-0.06} * P^{0.8} * T^{-0.4}(V_P + 1.4)^{0.8} \tag{11}$$

$$h_c = 7.799 * 10^{-3} * V_P^{\frac{1}{3}} * p^{0.5} * T^{0.5} \tag{12}$$

Equations (10)–(12) represent the coefficients of Woschni, Hohenberg and Eichelberg, respectively. $V_P$ is the average piston speed.

The term $W$ is deduced from Equation (13):

$$W = 2.28 * V_P + C_1 * \frac{V_d * T_a}{p_a * V_a}(p(\theta) - p_m) \tag{13}$$

with $C_1$ = 0.00324. On the other hand, $p_m$ represents the motored pressure, and subscript "$a$" is the reference condition

$$V_p = \frac{n * S}{30} \tag{14}$$

where $n$ is the engine speed (rpm).

## 2.1. Ignition Delay Model

The ignition of the fuel is not spontaneous; it is controlled by the Hardenberg and Haze model whose mathematical formulation is given by relation 14 [23–25]:

$$ID = (0.36 + 0.22\overline{V}_p)exp\left[E_A\left(\frac{1}{R\overline{T}} - \frac{1}{17.190}\right)\left(\frac{21.2}{\overline{p} - 12.4}\right)^{0.63}\right] \tag{15}$$

where $E_A$ is the activation energy, where, $R$ is the universal gas constant.

### 2.2. Détermining the Heat Release Rate

The net heat release rate (Net HRR) represents the thermal energy useful for engine operation [26–28]. It is transformed into mechanical energy by the piston during combustion inside the combustion chamber. The gross heat release rate (Gross HRR) is, in fact, the thermal energy transformed into mechanical work, and the energy losses in the walls are associated with it. The combustion of the fuel heats the cylinder walls by convection, and a finite amount of calorie is transmitted to the cooling water, constituting a heat loss. A curve representing the difference between the total combustion energy and the useful thermal energy is generated. The characteristic curves of conventional diesel D100 combustion are compared with those of biodiesel B100.

### 2.3. Experimental Setup

The experimental device is an air-cooled single-cylinder motor of the Lister Peter 0100529-TS1 series type, whose technical characteristics are confined to Table 1. It is a naturally aspirated, four-stroke engine with direct injection. Cooling is by ambient air. This equipment is located at Department of Energetic System and Sustainable Development (DSEE) of *Ecole des Mines de Nantes* (EMN) France. Figure 1 shows the experimental device used during the study.

**Table 1.** Engine characteristics.

| Lister-Petter-01005299-TS1 Série | | |
|---|---|---|
| Injection Pressure | (Bar) | 250 |
| Piston Diameter/Stroke | (mm) | 95.3/88.9 |
| Connecting Rod Length | (mm) | 165.3 |
| Engine Capacity | (m$^3$) | 630 |
| Compression Ratio | - | 18 |
| Injection Timing | (°CA) | 15° Before TDC |
| Engine Power | (kW) | 4.5 à 1500 trs/min |

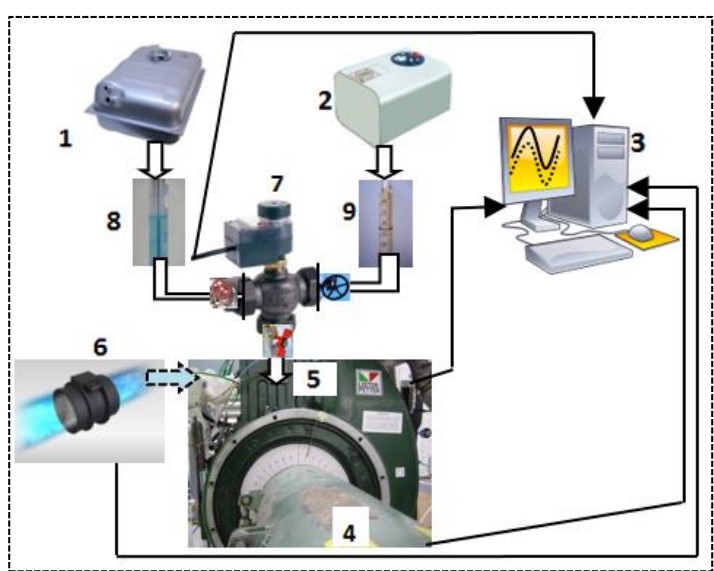

**Figure 1.** Experimental setup. Legend; 1. Diesel fuel tank; 2. Biodiesel tank; 3. High-frequency acquisition system and computer; 4. Brake dumb dynamometer; 5. Diesel monocylindric motor; 6. Debimeter and air flow sensor; 7. Three-way valve with a purge system; 8. Diesel burette; 9. Biodiesel burette.

The variation in the load was obtained by modifying the effective power of the engine by a dynamometer as well as the quantity of fuel admitted into the cylinders. The engine

load was varied at 25%, 50%, 75% and 100%. Effective power of 4.2 kW corresponds to the maximum engine load (100%).

The fuel supply system consists of two different fuel tanks, (1) and (2), that lead to the three-way valve (7). Each way of this valve was equipped with a stop valve. One tank was filled with conventional diesel fuel D100 (1) and the other (2) with the neem oil methyl ester biodiesel B100. The fuel change during the test was conducted in advance by the shutdown of the second fuel supply and the systematic purge by a system incorporated into the three-way valve. It took ten minutes, with the engine running, to begin measurements without the relative accidental risk of mixing the two fuels.

The acquisition of the cylinder pressure was carried out using the Indwin AVL engine rotating at 1500 rpm. The angular position of the crankshaft and the engine speed were obtained via an AVL 364C encoder, attached to the crankshaft. The encoder measures only parameters with frequencies greater than 90 kHz. Table 2 presents the test matrix table, which summarizes the main characteristics of the test carried out, and Table 3 presents the relative errors of the different sensors used.

**Table 2.** Main characteristics of the test.

| Fuels | Engine Speed rpm | Engine Load kW | | | | Injection Timing °CA |
|---|---|---|---|---|---|---|
| Diesel Biodiesel | 1500 | 1.18 | 2.13 | 3.38 | 4.59 | 15 |

**Table 3.** Main characteristics of the test.

| Parameters | Errors |
|---|---|
| Engine torque | ±0.1 N.m |
| Engine speed | ±3 rpm |
| Injection timing | ±0.05 °CA |
| Cylinder pressure | ±0.5 of the measured value |
| LCV | ±0.25% of the measured value |
| Admission air flow | ±0.1% of the measured value |
| Fuel flow | ±0.5% of the measured value |
| Injection pressure | ±2 bars |
| Inlet air temperature | ±1.6 °C |
| Exhaust air temperature | ±1.6 °C |

*2.4. Fuel Characteristics*

The experimentation resulting from the work of Ayissi et al. [29] was successfully repeated under the same conditions. The neem oil obtained was characterized in order to calibrate its properties and use it in a heat engine. The characteristic performance values obtained were mostly those found by the authors. These were adopted and are reported in Table 4.

**Table 4.** Fuel characteristics.

| Fuel Properties | (B100) | (D100) |
|---|---|---|
| Density (kg/m$^3$) | 883.3 | 830 |
| Cetane Number | 51.3 | 48 |
| Lower Heating Value LCV (MJ/kg) | 39.7 | 42.5 |

*2.5. Numerical Method*

Figure 2 presents a synoptic view of the overall methodological organization of the digital study. The end-of-compression characteristic parameters were considered and retained as well as the characteristics of the model engine. The intrinsic properties of the fuels were taken into account. A numerical code was developed. The characteristic equations of the implemented models were solved by the Runge–Kutta method.

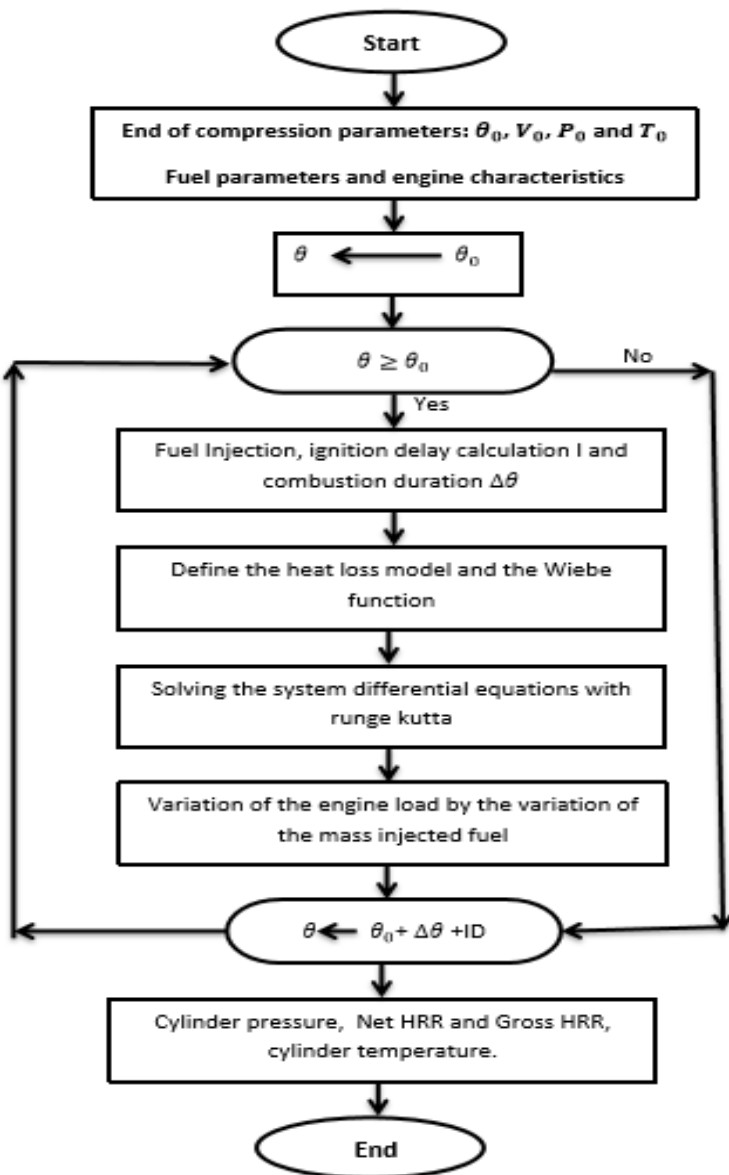

**Figure 2.** Digital diagram.

## 3. Results and Discussion

### 3.1. Experimental Evaluation of Cylinder Pressure

Figure 3 shows the increase in the experimental cylinder pressure based on the crankshaft angle when using B100 and D100 at 100% load. A similarity between the increase in the cylinder pressure curves of biodiesel B100 and D100 is observed. However, there is an estimated pressure increase of 3.3% when using B100 at 100% load. A faster pressure increase in B100 compared to D100 is also observed.

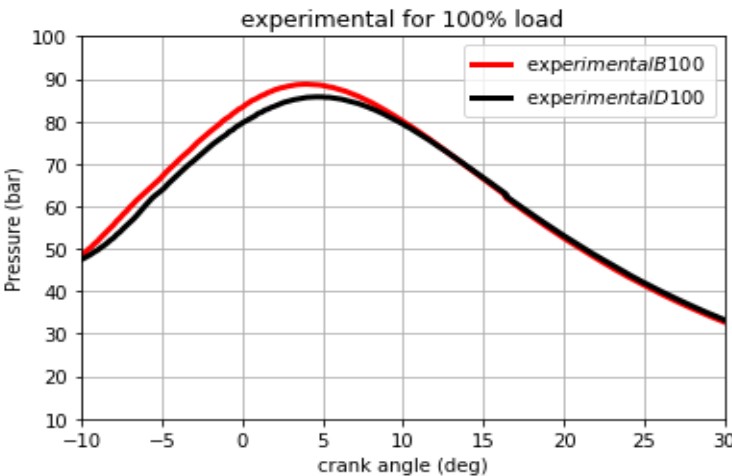

**Figure 3.** Comparison of experimental pressures of B100 and D100 at 100% load.

High cylinder pressure is characteristic of good fuel vaporization as well as relatively better oxygenation of fuel. This observation has also been made in the literature. Indeed, the studies of Tarabet [18], Mohamed et al. [24] and Evangelos [30] demonstrated this quite clearly. The transesterification process helps reduce the viscosity of biodiesel. The operation of the engine under high loads would sufficiently reduce the effects of the viscosity of biodiesel on the kinetics of combustion. The increase in cylinder pressure during biodiesel combustion is probably due to the relative simplicity of the molecular structure of the hydrocarbons that B100 contains. This molecular structure would be even more advantageous at high load since the high heat inside the cylinder would contribute to the destructuring of the macromolecules of biodiesel, which is relatively less complex after the transesterification process. This predisposition of biodiesel to good combustion could promote the rapid appearance of radicals inside the drops close to the stoichiometry and catalyzed by the high cylinder temperature. This spatial–temporal consideration would suit the premix combustion process with the consequence of the increase in the pressure peak in the cylinders. The intramolecular presence of residual oxygen atoms would be another factor in raising the pressure peak in the cylinders as biodiesel is relatively oxygenated and the kinetics of combustion are easy. This high-pressure peak would be one of the reasons that increases the NOx level in post-combustion gases of an engine fueled with biodiesel generally [5,6,26]. Chiavola et al. [22] and Tarabet [18] mentioned it. The faster pressure increase in B100 biodiesel compared to diesel is characteristic of a short ignition period. Mohamed et al. and Evangelos [20,21,27] made the same observation in similar studies.

### 3.2. Cylinder Pressure Evaluation Considering the Heat Loss through the Cylinder Wall Patterns of B100 and D100 at 100% Load

Figures 4 and 5 show the evolution of the cylinder pressure during the combustion of B100 and D100, respectively. Under the constraint of three heat loss models, namely, Eichelberg, Hohenberg and Woschni, the time evolution of the cylinder pressure is simulated taking into account the heat loss through the cylinder wall and associated with the experimental pressure curve. A relative similarity between the numerical and experimental pressure curves for both B100 and D100 combustions is observed. The peaks characteristic of the temporal variation in the cylinder pressure of the three numerical models are evaluated at 90, 92 and 85 bars, respectively, for the Eichelberg, Hohenberg and Woschni models. The result of the experimental set is 89 bars. The overestimation of cylinder pressures by the implemented models compared to the experimental study is explained by the fact that the numerical models take into account the heat loss through the cylinder wall. The difference observed in the variations in the pressure peaks of the implemented models, although varying from one correlation to another, remains confined to the intervals encountered in the literature.

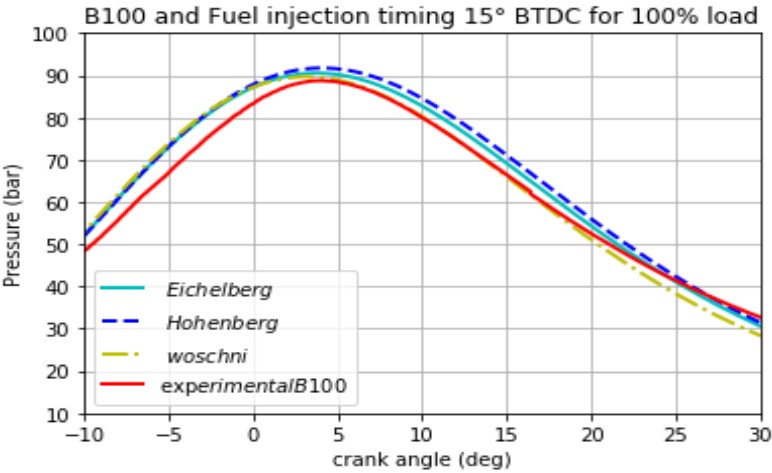

**Figure 4.** Cylinder pressure of B100 at 100%.

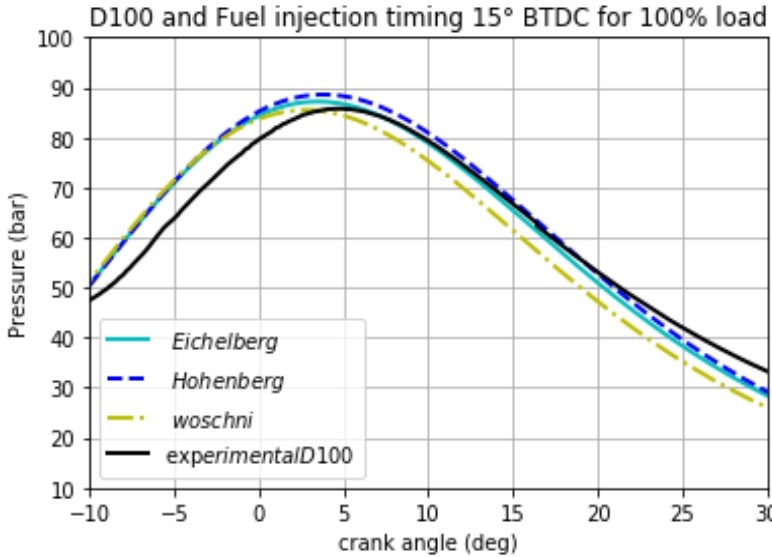

**Figure 5.** Cylinder pressure of D100 at 100%.

The three wall loss models studied represent fairly well the combustion of the model fuels in the compression-ignition configuration.

In the specific case of B100 combustion illustrated in Figure 4 at 100% load, the characteristic curves of the three models studied keep the same spatial position during the initial 12 °CA of combustion. Between the beginning of the rise in cylinder pressure and its peak, a gap of 7 bars is observed between the cumulative numerical curves and the experimental curve. This gap which benefits the simulated correlations is probably due to the fact that the simulated models take into account the heat loss through the cylinder wall in the physical materialization of the spatiotemporal evolution of the cylinder pressure.

The cylinder pressure of the Woshni model begins a declination around 12 °CA after the start of the rise in the cylinder pressure. The initiated pressure drop falls below the characteristic values of the experimental cylinder pressure. This underestimation of the time evolution of the cylinder pressure by the correlation proposed by Woshni could be assimilated by the fact that it does not take into account the piston speed. The piston displacement would present the cylinder surface at the temperature gradient generated by combustion, which would more or less increase the heat loss through the cylinder wall. Hohenberg's model is the one that retains the highest pressure peak of all the correlations implemented.

In the specific case of Figure 5, it is observed that the curves of the numerical models remain confused during the first 10 °CA before seeing the declination in the characteristic curve of the Woshni model. At this load range, all the models shown decline below the experimental cylinder pressure around 17 °CA. Heat losses at 75% load could be minimized in the post-combustion period. A similar observation is made at 100% load, under the same simulation conditions, although the declination occurs at 25 °CA.

### 3.3. Cylinder Pressure Evaluation Taking into Account the Heat Loss through the Cylinder Wall Patterns When Engine Is Fueled with B100 at 25%, 50%, 75% and 100% Load

Figures 6–8 show the evolution of the cylinder pressure at 75%, 50% and 25% load, respectively. A decrease in the cylinder pressure proportional to the load is observed at all measuring points. The decrease in the mass of fuel allowed would be at the origin of this observation during the combustion of B100.

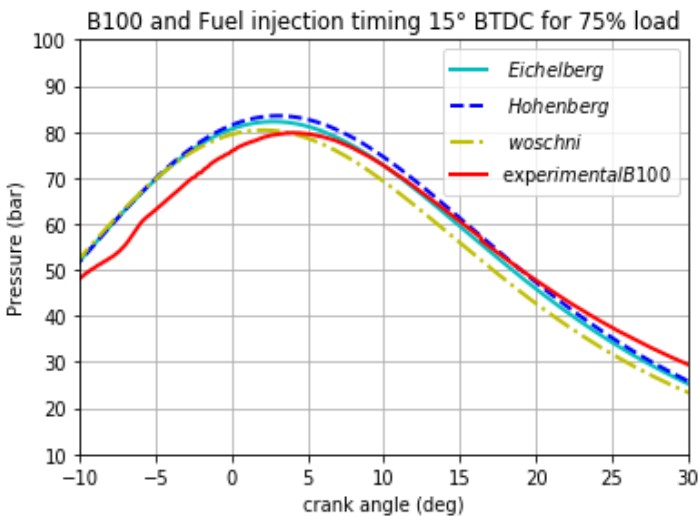

**Figure 6.** Cylinder pressure of B100 at 75% load.

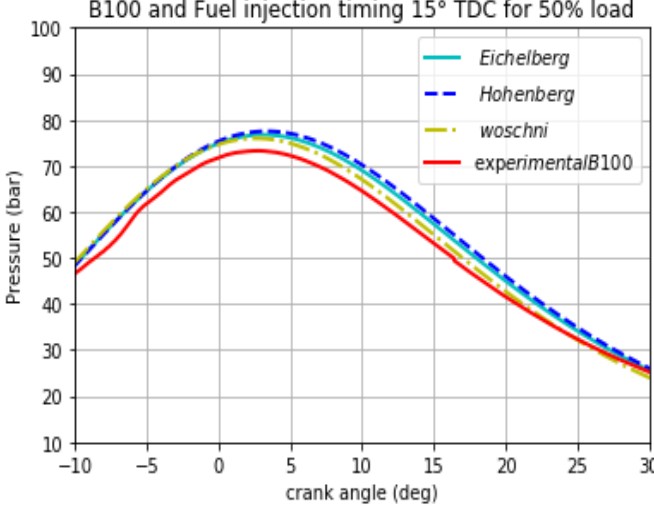

**Figure 7.** Cylinder pressure of B100 at 50% load.

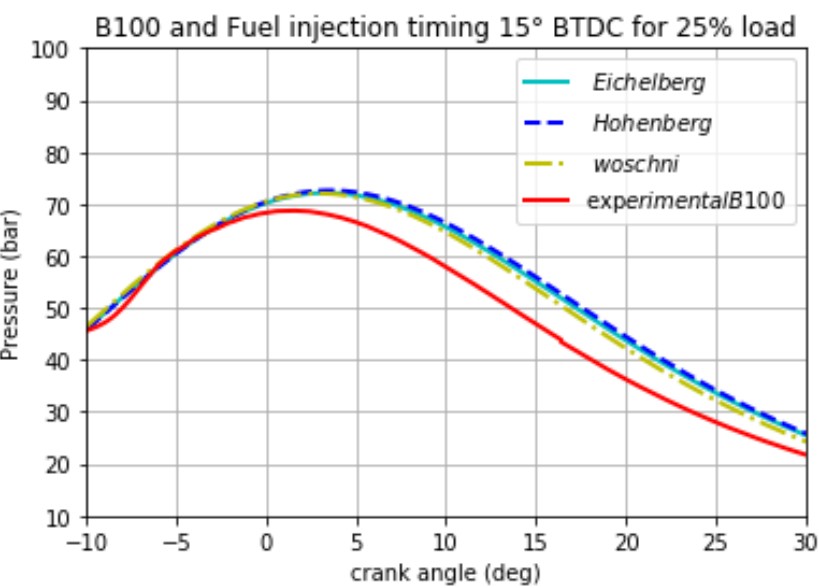

**Figure 8.** Cylinder pressure of B100 at 25% load.

The experimental pressure decreases by 10% and 6% proportionally to the passage from 100% to 75% load and from 75% to 25% load successively. This drop in experimental pressure is evaluated at 22.5% between 100% and 25% load. This decrease in cylinder pressure could be attributed to the reduction in effective power characterized by the decrease in the quantity of fuel admitted into the cylinders. However, with regard to the evolution of the pressure curves whatever the type of model implemented, the thermal losses would be more important during the ascending phase of the piston.

Considering the specific combustion of B100 at 75% load, the statistical differences between the simulated pressure curves and the experimental curve, between 20 and 30 °CA, are the smallest, all loads combined. This observation visible in Figure 6 would indicate that the heat losses are less important at this phase of engine operation. The rising part of the numerical pressure curves shows the largest deviations from other load points studied. At this load range, the Woschni model underestimates the approximation of the cylinder pressure value.

The 50% load range seems to present the best compromise between the estimated values of wall losses according to the models studied. This load range would be relatively close to the nominal operating rating of the test engine.

Indeed, when compared with other load ranges studied, a better compromise is observed between the evolutions of the combustion models and the experimental curve. The deviations observed during the other load ranges evaluated remain minimized for the Woschni, Eichelberg and Hohenberg correlations.

At 25% load, the three models have curves representing the temporal evolution of the cylinder pressure which are in agreement. This observation can be seen more clearly in Figure 8. This observation could be explained by a small parametric difference between the values of the different correlations implemented.

In general, the Eichelberg and Hohenberg models agree on the time evolution of the cylinder pressure during the tests at 50% and 25% load. This rise in digital cylinder pressure is characteristic of the rise deduced from the heat loss through the cylinder wall during the descending phase of the pressure curves considered. The tendency for heat loss to increase during this phase and at this load range could be due to the fact that the gases have more time to lick the walls because of the reduced speed of the piston in its descending phase.

Observation at high loads shows that the Eichelberg and Hohenberg models show a slight difference in the approximation of the cylinder pressure evolution. However,

this characteristic deviation remains less than 0.6 bar regardless of the time position of the piston. The Woschni model seems to underestimate the time value of the heat loss through the cylinder wall regardless of the engine load point compared to the other two correlations implemented.

### 3.4. Evaluation of Heat Release

The heat release at different engine loads at 100%, 75% and 50% is evaluated. The heat release curves of the different fuel models D100 and B100 have characteristic curves of combustion by auto-ignition.

Whatever the load range or the fuel tested, the evolution profiles of net heat release rate (Net HRR) and gross heat release rate (Gross HRR) during the premix pressure rise phase remain the same. A relatively significant difference is observed between Net HRR and Gross HRR when compared with D100 diesel. These differences are evaluated at 10, 10, and 20 J/°CA for 50%, 75% and 100% load, respectively. These differences would mean that heat losses would be greater during combustion of B100 biodiesel. Heat losses in the premix phase would be minimized based on information from the study.

### 3.4.1. Heat Release Curve for 100% Load

Figures 9 and 10 show the different heat release profiles of D100 diesel and B100 biodiesel, respectively, for a 100% load range, equivalent to 4.5 kW. The Net HRR converted by the engine to mechanical energy when B100 is used as a test fuel is approximately 30 J/°CA. Between 0 and 10 °CA, this biofuel provides an estimated energy of 17.32 J/°CA compared to 21.31 J/°CA for D100. No fundamental difference between the different Gross HRR values of the two fuels over this load range is observed. B100 biodiesel would have better performance at high loads. This behavior is probably due to the oxygenation of the molecule and the structural simplicity of its chemical composition [23–26,30]. At high temperatures, the relatively higher viscosity of biodiesel is no longer fundamentally a handicap for the combustion of this fuel. This would justify its relatively good combustion.

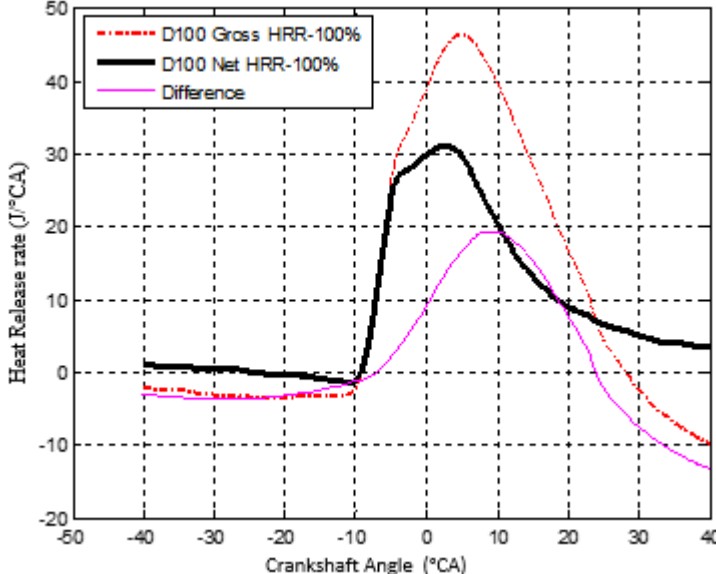

**Figure 9.** Profile of net heat release rate (Net HRR) evolution and gross heat release rate (Gross HRR) for engine speed n = 1500 rpm at 100% load according to D100 combustion.

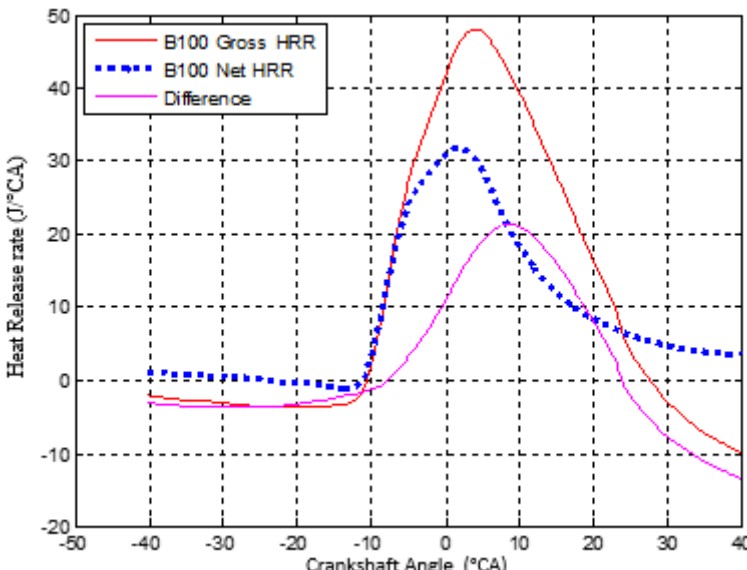

**Figure 10.** Profile of net heat release rate (Net HRR) evolution and gross heat release rate (Gross HRR) for engine speed n = 1500 rpm at 100% load according to B100 combustion.

### 3.4.2. Heat Release Curves Net HRR and Gross HRR for 75% Load

Figures 11 and 12 show the different profiles the net heat release rate (Net HRR) and the gross heat release rate (Gross HRR) for an engine speed of 1500 rpm at 75% load. Between 0 and 10 °CA, the combustions of B100 and D100 produce 13.20 and 14.20 J/°CA, respectively. The heat loss through the cylinder wall as a function of the crankshaft position is evaluated at about 10 J/°CA during B100 combustion, representing 20% higher than during the combustion of D100 at this range of load. A similar observation was made by Pankaj et al. [31] and Muhamed et al. [28]. These authors showed that at a range of 75%, the ignition delay of B100 and D100 increases. This increase of about 2 °CA in the ignition delay is significant and has also been observed by Muhamed et al. [27] at this range of load.

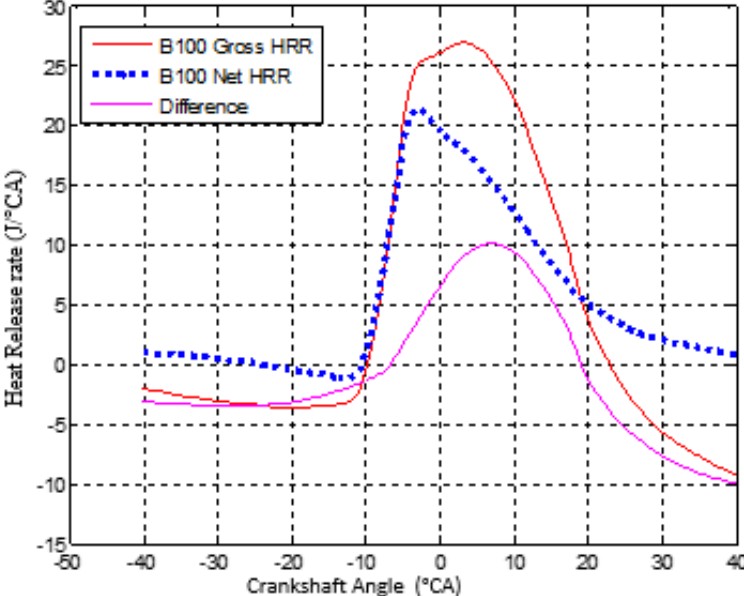

**Figure 11.** Profile of net heat release rate (Net HRR) evolution and gross heat release rate (Gross HRR) for engine speed n = 1500 rpm at 75% load according to the B100 combustion analysis model.

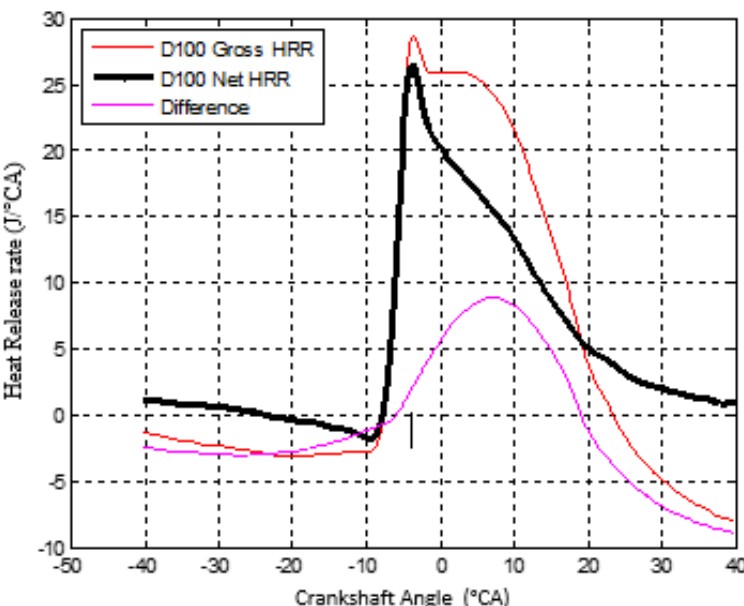

**Figure 12.** Profile of net heat release rate (Net HRR) evolution and gross heat release rate (Gross HRR) for engine speed n = 1500 rpm at 75% load according to D100 combustion.

### 3.4.3. The Net HRR and Gross HRR for 50% Load

Figures 13 and 14 show the different profiles of the evolution of the net heat release rate (Net HRR) and the gross heat release rate (Gross HRR) for an engine speed of 1500 rpm at 50% load. These two figures indicate that the diffusion combustion phase of B100 and D100 starts with a Net HRR of 20 and 21 J/°CA, respectively. The heat loss through the cylinder wall (curve representing the difference) is estimated at 10 for B100 and 8 J/°CA for D100. At 10 °CA, the Net HRR value of B100 is 12 and 13.5 J/°CA for D100. The heat loss through the cylinder wall values are as follows: B100 (10 J/°CA), D100 (8 J/°CA).

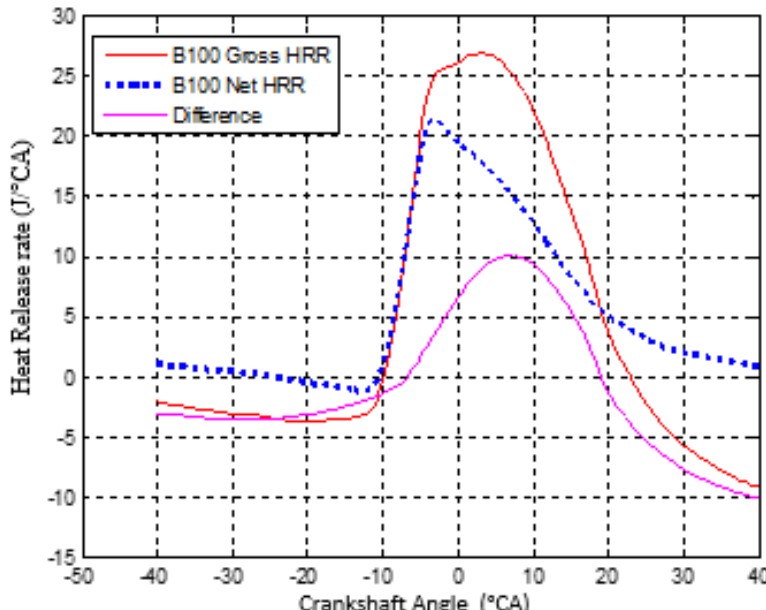

**Figure 13.** Profile of net heat release rate (Net HRR) evolution and gross heat release rate (Gross HRR) for engine speed n = 1500 rpm at 50% load according to B100 combustion.

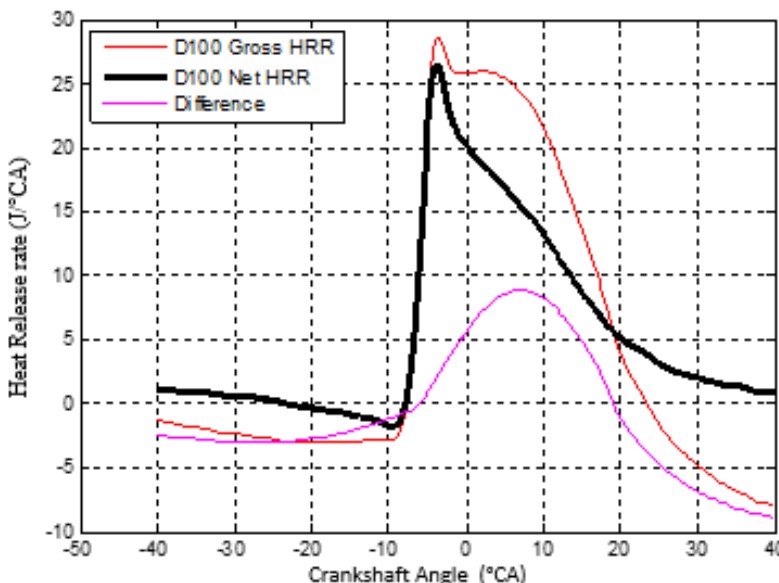

**Figure 14.** Profile of net heat release rate (Net HRR) evolution and gross heat release rate (Gross HRR) for engine speed n = 1500 rpm at 50% load according to D100 combustion.

## 4. Conclusions

An experimental investigation coupled with a thermodynamic 0D simulation of a neem-based biodiesel, a conventional diesel and their respective different heat releases was carried out. The different correlations of Woschni, Eichelberg and Hohenberg served as numerical models in the systemic simulation of combustion phenomenology. The Woshni correlation best represents the peak of the cylinder pressure comparison made with other implemented models. This correlation, however, seems to relatively underestimate the value of the cylinder pressure after bottom dead center. The Eichelberg and Hohenberg models represent the spatial–temporal evolution of the cylinder pressure in the same way. Implementation of other heat losses through cylinder wall models such as Zainal or Sitkey could be exploited for greater openness to the field of investigation. The investigation of the cylinder pressure confirmed that the load rate has an influence on the value of the peak of the cylinder pressure. The study presented provides a better understanding of the phenomenology of parietal thermal losses from the combustion of alternative fuels and its impact on the performance of combustion ignition engines. The cylinder pressure in the case of the combustion of biodiesel B100 is estimated at 89 bars against 86 bars for diesel D100; a gap of 3.3% to the advantage of B100 is thus observed when using this biofuel at 100% load. A faster pressure increase in B100 compared to D100 is also observed. A study of a parametric variation in the ignition delay associated with this study could better explain its influence on the comparative evolution of the peak cylinder pressure depending on the nature of the fuel.

**Author Contributions:** Conceptualization, D.R.K.L. and Z.M.A.; methodology, M.O.; software, D.R.K.L. and Z.M.A. and M.H.B.; validation, E.F.H.P., M.O. and Z.M.A.; formal analysis, Z.M.A.; investigation, Z.M.A. and D.R.K.L.; resources, M.O.; data curation, D.R.K.L.; writing—original draft preparation, Z.M.A. and D.R.K.L.; writing—review and editing, Z.M.A. and D.R.K.L.; visualization, M.H.B.; supervision, E.F.H.P.; project administration, M.O.; funding acquisition, Z.M.A. and D.R.K.L. All authors have read and agreed to the published version of the manuscript.

**Funding:** This research received no external funding.

**Institutional Review Board Statement:** Ethical review and approval were waived for this study, due to the fact that the studies not involve humans or animals.

**Informed Consent Statement:** Patient consent was waived due to the reason that the studies not involving humans.

**Acknowledgments:** We thank the Department of Energetic System and Sustainable Development of *Ecole des Mines de Nantes* EMN for giving us the experimental device; the G.C.C. team for always encouraging us in combustion and simulation aspects; and the team members of Energy, Electric and Electronic Systems, University of Yaoundé 1, for welcoming and assisting us.

**Conflicts of Interest:** The authors declare that there is no conflict of interests regarding the publication of this paper.

## Abbreviations

| | |
|---|---|
| $\theta$ | Crank angle (°CA) |
| $\Delta\theta$ | Total combustion duration (°CA) |
| $\theta_0$ | Start of combustion (°CA) |
| $x$ | Burnt mass fraction |
| $W$ | Work done (KJ) |
| $U$ | Internal energy |
| $CR$ | Compression ratio |
| $c_v$ | Specific heat at constant volume |
| $h_c$ | Heat transfer coefficient (W. $m^{-2}.K^{-1}$) |
| $m$ | Masse (Kg) |
| $R$ | Gas constant |
| $R$ | Gas constant |
| $Q$ | Heat transfer (KJ) |
| $V$ | Volume ($m^3$) |
| $p$ | Pressure (Bar) |
| $T$ | Temperature (K) |
| $A(\theta)$ | Area exposed to heat transfer ($m^2$) |
| $D$ | Cylinder bore (m) |
| S | Stroke (m) |
| $n$ | Engine speed |
| $\omega$ | Angular velocity (rad/s) |
| $V_p$ | Mean piston speed (rad/s) |
| TDC | Top dead center (deg) |
| BTDC | Before TDC (deg) |
| BDC | Bottom dead center (deg) |
| HRR | Heat release rate |
| °CA | Degree crank angle |

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
