# Peer review of "Experimental and Simulation of Diesel Engine Fueled with Biodiesel with Variations in Heat Loss Model"

_energies, doi:10.3390/en14061622_

Round 1

Reviewer 1 Report

This is an experimental study on a single cylinder Diesel Engine running on 100% biodiesel, compared to normal Diesel fuel. Indicator diagrams are recorded for 1500 rpm engine speed and 25, 50, 75 and 100% load. Heat release rates are computed based on the analysis of indicator diagrams, based on three alternative engine cooling losses models. An attempt is made to optimize injection timing for operation with the biodiesel fuel.

The authors have done a systematic experimental – and computational – effort. However, they must significantly improve their presentation.

The experimental section needs to be enhanced with a test matrix table, which will summarize the tests carried out with their main characteristics (engine speed, load, fuel type, injection timing). Another Table is necessary with the sensors and instruments employed in the measurements, with the respective accuracy. Data acquisition details should also be included.

Line 25 and other places: avoid the word “parietal”, it is not appropriate for engine heat losses.

Line 27: “It also shows that the problem of high pressure related to the use of biodiesels in engines can be solved by varying the engine load”. This statement makes no sense. Do you mean “…by optimizing injection timing” ?

English needs significant improvement and check by a native speaker.

Author Response

The experimental section needs to be enhanced with a test matrix table, which will summarize the tests carried out with their main characteristics (engine speed, load, fuel type, injection timing). Another Table is necessary with the sensors and instruments employed in the measurements, with the respective accuracy. Data acquisition details should also be included.

Table 2 presents the test matrix table, which summarize the main characteristics of the test carried out.

Fuel type

Engine speed (trs/min)

Engine load (kW)

Injection timing (°CA)

Diesel

1500

1,18

2,13

3,38

4,59

15

Methyl Ester of Neem Biodiesel

Table 1 presents the relative errors of the different sensors used

Parameters

Errors

Engine torque

± 0,1 N.m

Engine speed

± 3 tr/min

Injection timing

±0,05 °CA

Cylinder Pression

± 0,5 of the measured value

Lower Heating Value LCV

± 0,25 % of the measured value

Admission air flow

± 0,1 % of the measured value

Fuel Flow

± 0,5 % of the measured value

Injection Pressure

± 2 bars

Inlet air temperature

±1,6 °C

Exhaust air temperature

±1,6 °C

Line 25 and other places: avoid the word “parietal”, it is not appropriate for engine heat losses.

« parietal » has been changed by « walls » accordingly throughout the text

Line 27: “It also shows that the problem of high pressure related to the use of biodiesels in engines can be solved by varying the engine load”. This statement makes no sense. Do you mean “…by optimizing injection timing” ?

« …varying the engine load » has been changed by « optimizing injection timing » accordingly

English needs significant improvement and check by a native speaker.

The English has been revised throughout the text

Reviewer 2 Report

- Line 89. Incomprehensible term „Vilebrequin“.

- Line 118. Incomprehensible terms „Avec“, „D'autre part“.

- Equation (3) and elsewhere. The pressure in the cylinder must be marked not P (uppercase letter) but p (lowercase letter). Rotational speed of the engine (n) must be marked not uppercase letter, but lowercase letter.

- Line 387. Incorrect units Sp (rad/s). Line 118. Mean piston speed should be denoted not Sp but Vp.

- Many of the equations in the article are presented without citing other authors. Did the authors of this article really create the equations provided?

- All elements of the equations must be explained.

- Figure 1 must be of higher quality. The internal combustion engine in the illustration must be mounted on the engine load bench.

- Please describe the engine fueling system.

- A description of the recording in the cylinder pressure recording device must be provided when describing the experimental equipment.

- Line 226, 228 and elsewhere. The notation °V is used, which is incomprehensible. There must be an explanation.

- The conclusions should be more specific and accurate, substantiated by the study conducted.

Author Response

- Line 226, 228 and elsewhere. The notation °V is used, which is incomprehensible. There must be an explanation.

« °V » has been changed by cranshaft angle degree « °CA »

- Line 118. Incomprehensible terms „Avec“, „D'autre part“.

« Avec » has been changed by « With » ; « d’autre part » has been changed by « On the other hand »

-Equation (3) and elsewhere. The pressure in the cylinder must be marked not P (uppercase letter) but p (lowercase letter). Rotational speed of the engine (n) must be marked not uppercase letter, but lowercase letter.

« P » has been changed by «p» «N» has been changed by «n»

- Line 387. Incorrect units Sp (rad/s). Line 118. Mean piston speed should be denoted not Sp but Vp.

« Sp » has been changed by «p» « Vp »

- All elements of the equations must be explained.

All elements of the equations are now explained

- Figure 1 must be of higher quality. The internal combustion engine in the illustration must be mounted on the engine load bench.

The quality of figure 1 has been amelioreted and load bench mounted on the engine accordingly

- Please describe the engine fueling system.

2.5. Experimental Set-Up

Figure 1, shows the experimental set up used in the study. The experimental device is an air-cooled single-cylinder motor of the Lister Peter 0100529-TS1 series type, whose technical characteristics are confined to Table 1. It is a naturally aspirated, four-stroke en-gine with direct injection. Cooling is by ambient air. Figure 1 shows the experimental de-vice used during the study.

- The conclusions should be more specific and accurate, substantiated by the study conducted.

An experimental investigation coupled with a thermodynamic zero-dimention simulation of a neem-based biodiesel, a conventional diesel and their respective different heat releases was carried out. The different correlations of Woschni, Eichelberg and Hohenberg served as a numerical model in the systemic simulation of combustion phenomenology. woshni correlation best represents the peak of the cylinder presion comparison made with other implemented models. this correlation, however, seems to relatively underestimate the value of the cylinder pressure after Botom dead center. The Eichelberg and Hohenberg model represent the spatial-temporal evolution of cylinder pressure in the same way. Implementation of other wall heat losses such as Zainal or Sitkey could be exploited for greater openness to the field of investigation. The investigation of the cylinder pressure confirmed that the load rate has an influence on the value of the peak of the cylinder pressure. The study presented provides a better understanding of the phenomenology of parietal thermal losses from the combustion of alternative fuels and its impact on the performance of combustion ignition engines. The cylinder pressure in the case of the combustion of biodiesel B100 is estimated at 89 bars against 86 bars for diesel D100 a gap of 3.3% to the advantage of B100 is thus observed when using B100 at 100% load. A faster pressure increase of B100 compared to D100 is also observed.

Reviewer 3 Report

The manuscript deals with a numerical and experimental investigation of a diesel engine fuelled with biodiesel at different engine load and employing different heat loss model.

The authors should improve the readability and scientific soundness, the manuscript cannot be accepted in this present form, please carefully revise it to improve the quality, the reviewer has been highlighted several open points below:

Please summarize the title is too long, maximum 15 words could be sufficient. For example, Experimental and 0D simulation of Diesel Engine fuelled with biodiesel.

Reading the abstract, it is difficult to understand the novelty of this manuscript, please clarify the main message of your paper. Several published papers have reported the sensitivity between biodiesel and engine load compared to the conventional diesel fuel. Then, the authors have reported that with biodiesel higher peak cylinder pressure, 3 bar more than diesel, please clarify the physics beyond.

The bibliographic review performed in the introductory section is very well structured but does not cover all the progress and actual state of the art of the Compression Ignition engines and as well as alternative fuels, the reviewer suggestion is to take a look at the work performed at the Istituto Motori (look for Dr Di Blasio) and Sandia (look for Dr Mueller). It is worth to mention their contribution. They provide a wider overview of the combustion technologies that have been tested able to improve the CO2 and the NOx-Soot trade-offs such as specific bowl design, innovative fuel injection systems.

Please avoid lump sum references such as “0D models of combustion are widely used in the literature [8-19].”. Each reference should be described in detail, discriminating the main outcomes among all.

Between number and dimension unit please add a space, check carefully all the manuscript., such as "1.1kW, 2.5kW, 3.3kW and 4.5kW, respectively."

Please revise carefully all figures and plots, the readability should be improved. Figure 9's quality and readability are very low.

Table 2, please revise the dimension units, there is a typo Density should be expressed in kg/m3 instead of Kg/m3, revise carefully all the manuscript.

Novelty is not clear, please emphasise it, why are fundamental the outcomes reported in this manuscript for the literature?

I recommend that you edit the entire article according to journal standards and re-send it. An article that is not well prepared, it is more a research technical report, I regret to inform you that it cannot be accepted for publication in a highly prestigious journal like MPDI.

Author Response

Please summarize the title is too long, maximum 15 words could be sufficient. For example, Experimental and 0D simulation of Diesel Engine fuelled with biodiesel.

Title is summarized

“Experimental and simulation of Diesel Engine fuelled with biodiesel with variations in heat loss model”

Reading the abstract, it is difficult to understand the novelty of this manuscript, please clarify the main message of your paper. Several published papers have reported the sensitivity between biodiesel and engine load compared to the conventional diesel fuel. Then, the authors have reported that with biodiesel higher peak cylinder pressure, 3 bar more than diesel, please clarify the physics beyond.

3.1. Experimental Evaluation of Cylinder Pressure

Figure 3 shows the increase of experimental cylinder pressure based on crankshaft angle when using B100 and D100 at 100% load. A similarity between the increase of cylinder pressure curves of biodiesel B100 and D100 is observed. However, there is an estimated pressure increase of 3.3% when using B100 at 100% load. A faster pressure increase of B100 compared to D100 is olso observed.

High cylinder pressure is characteristic of good fuel vaporization as well as relatively bet-ter oxygenation of fuel. This observation has also been made in the literature. Indeed, the studies of Tarabet [23], Mohamed et al [27] and Evangelos [29] demonstrate this quite clearly. The transesterification process helps reduce the viscosity of biodiesel. The operation of the engine under high loads, would sufficiently reduce the effects of the viscosity of biodiesel on the kinetics of combustion. The increase in cylinder pressure during biodiesel combustion is probably due to the relative simplicity of the molecular structure of the hy-drocarbons of B100 contain. This molecular structure would be even more advantageous at high load since the high heat inside the cylinder would contribute to the destructure of the macromolecules of biodiesel, relatively less complex after the transesterification process. This predisposition of biodiesel to good combustion, could promote the rapid appearance of radicals inside the drops close to the stechiometry and catalyzed by the high cylinder temperature. This spatial-temporal consideration would suit the premix combustion process with the consequence, the increase in the pressure peak in the cylinders. The intramolecular presence of residual oxygen atoms would be another factor in raising the pressure peak in the cylinders as biodiesel is relatively oxygenated and the kinetics of combustion being easy. This high pressure peak would be one of the reasons that increase the NOx level in post-combustion gases of engine fuelled with biodiesel generaly [5, 6, 30]. Chiavola et al [19] and Tarabet [23] mention it. A faster pressure increase of B100 bio-diesel compared to diesel is characteristic of a short ignition period. Mohamed et al and Evangelos [24,25,31] have made the same observation in similar studies.

The bibliographic review performed in the introductory section is very well structured but does not cover all the progress and actual state of the art of the Compression Ignition engines and as well as alternative fuels, the reviewer suggestion is to take a look at the work performed at the Istituto Motori (look for Dr Di Blasio) and Sandia (look for Dr Mueller). It is worth to mention their contribution. They provide a wider overview of the combustion technologies that have been tested able to improve the CO2 and the NOx-Soot trade-offs such as specific bowl design, innovative fuel injection systems.

We have add news references:

The combustion of diesel engines offers a wide area of investigation [4,14,18]. Experimental work is being carried out. The Ducted fuel injection (DFI) is proposed in the literature as a strategy to improve the fuel/gas load mixture of the compression ignition engine relative to conventional diesel combustion (CDC) [5-7]. The concept of DFI is to inject each fuel spray through a small tube into the combustion chamber to facilitate the creation of a leaner mixture in the self-ignition zone, compared to a fuel jet without being sur-rounded by a duct. The experiments are interesting. Nilsen et al. [5] studies the effects on emissions and engine efficiency using a two-hole injector tip for charging gas mixtures containing 16 and 21% mol oxygen. DFI seems to confirm that it is effective in reducing engine soot emissions. Soot and NOx are being reduced with increasing dilution. Christopher [6] conduct another experience. DFI and CDC was directly compared at each operating point in the study. At the low-load condition, the intake charge dilution was swept to elucidate the soot and NOx

Round 2

Reviewer 1 Report

The authors could further improve the quality of their presentation along the guidelines set in my first review.

Nevertheless, the manuscript is publishable in its revised form.

Reviewer 2 Report

The quality of Figure 1 should be improved.

Reviewer 3 Report

The manuscript has been revised, I would like to recommend it for publication.